# Cu(II)-Catalyzed C-N Coupling of (Hetero)aryl Halides and *N*-Nucleophiles Promoted by *α*-Benzoin Oxime

**DOI:** 10.3390/molecules24224177

**Published:** 2019-11-18

**Authors:** Chunling Yuan, Lei Zhang, Yingdai Zhao

**Affiliations:** Department of Medicinal Chemistry, Pharmacy School, Jinzhou Medical University, Jinzhou 121001, China; hm7835831@163.com (L.Z.); ZYd1216221530@163.com (Y.Z.)

**Keywords:** α-Benzoin oxime, Copper catalyst, (Hetero)aryl halides, *N*-Nucleophiles, C-N Coupling

## Abstract

We first reported the new application of a translate metal chelating ligand α-benzoin oxime for improving Cu-catalyzed C-N coupling reactions. The system could catalyse coupling reactions of (hetero)aryl halides with a wide of nucleophiles (e.g., azoles, piperidine, pyrrolidine and amino acids) in moderate to excellent yields. The protocol allows rapid access to the most common scaffolds found in FDA-approved pharmaceuticals.

## 1. Introduction

*N*-Arylated compounds are ubiquitous synthons in numerous natural products and functional molecules [1,2]. Particularly, their most important function is as structural fragment for FDA-approved pharmaceuticals (Figure 1), for example, antibacterial agents (Ofloxacin [3] and Pipemidic acid [4]), antidiabetic drug (Repaglinide [5]), nonsteroidal anti-inflammatory drug (Celecoxib [6]), antihypertensive drug (Prazosin [7]), etc. Although C-N bond-formation reactions between (hetero)aryl halides and *N*-nucleophiles are well known, including the S_N_Ar reactions [8], classical Ullmann-type coupling reactions [9,10,11], and Buchwald-Hartwig reactions [12,13,14]. Amongst Pd-catalyzed C-N coupling reaction has been confirmed as a very useful method to build aromatic amines or arylazoles [15,16,17,18]. However, considering to the cost and toxicity of both palladium catalyst and auxiliary phosphine ligands, the arylation of *N*-nucleophiles with (hetero)aryl halides to form new C-N bonds still remains a significant opportunity. In recent years, Cu-catalyzed Ullmann-type couplings have attracted more and more attention due to the inexpensive catalyst and low toxicity. Although many commercially available or novel designed ligands have been developed for promoting copper-catalyzed couplings of aryl halides with *N*-nucleophiles, few of them are effective with less inactivated heterocyclic aryl chlorides. Besides, an easy-removal catalyst system is also very important for the environment.

During the past few years, Ma and co-workers have reported that the oxalic diamide ligands are powerful ligands for the copper-catalyzed couplings [19,20,21]. In the presence of ligands, the reaction temperature and catalyst loading could be significantly decreased while the yields were increased [22,23,24]. Following these studies, a number of bidentate ligands were reported for the synthesis of *N*-arylated compounds.

On the other hand, α-benzoin oxime (BO) is a common ligand, which is usually applied to inspect and measure copper, molybdenum, and tungsten [25,26], and also as a chelating agent for extracting antimony, vanadium, tungsten [27]. However, the use of BO in improving the Cu-catalyzed Ullmann style reactions is never reported. Herein, it was found that BO could be used in the direct couplings of the (hetero)aryl halides with *N*-nucleophiles. The reaction allowed rapid access to *N*-arylated compounds, the most common scaffolds found in FDA-approved pharmaceuticals. These reactions were occurred at mild temperature (80 °C), with employing (hetero)aryl halides, nucleophiles (e.g., azoles, piperidine, pyrrolidine and amino acids) and inexpensive catalysts, and affording high yields. Importantly, this process was general with respect to both the (hetero)aryl halides and nucleophiles, including the use of secondary amines and amino acids.

## 2. Results and Discussions

To initiate our studies, 2-bromoanisole was treated with pyrrole in the presence of 0.28 mol CuI in DMF. Regrettably, coupling product **3a** was obtained in 12% yield, along with unreacted starting material (Table 1, entry 1). The use of Cu powder and Cu(OTf)_2_ as catalyst in DMF delivered **3a** in low yields (entries 2, 3). Consistently, Cu(II) gluconate and Cu_2_(OH)_2_CO_3_ did not provide the desired product (entries 4, 5), only a trace of **3a** was observed. In the event, 10 mol% Cu(OAc)_2_ enabled the coupling of 2-bromoanisole and pyrrole to provide the desired product in 53% yield (entry 6). Attempts to improve the yield through changing the base were successful, with K_3_PO_4_ proving to be optimal in terms of yield (entries 6–11). Furthermore, with dioxane, toluene, DCE, H_2_O as solvents (entries 12–16), the yield was lower than that in DMF. Fortunately, DMSO gave significantly better results, and **3a** was obtained in a much-improved 90% yield. Meanwhile, lowering the temperature to 60 °C decreased the yield and led incomplete conversion (entry 17). A significant amount of coupling product was observed when using BO as ligand, comparing with the corresponding control experiments (entries 12 and 18). The scalability of this protocol was also tested through synthesis of the **3a** on a 4.4 g scale (entry 19). Additionally, the BO-promoted C-N reaction was also carried out under air, **3a** was obtained in 61% yield, indicating that this catalytic system was sensitive with air (entry 20). Finally, we checked other ligands (including *N*,*N*-, *N*,*O*-, *O*,*O*-type bidentate ligands) that were applied in Cu-catalyzed *N*-arylation reactions, and discovered that only **L3** and **L4** could furnish **3a** in moderate yields under standard condition. It was reasoned that these reported ligands may be effective when chelating with Cu(I).

With optimized conditions in hand, we set out to evaluate the scope of (hetero)aryl halides that would participate in this transformation (Table 2). The reaction tolerated a diverse array of functional groups on the (hetero)aryl halides, including methoxy (**3a**), aldehyde (**3c**), carboxyl (**3d**), amino (**3e**), ketone (**3f**), ester (**3g**, **3h**), cyano (**3i**). Electronic properties of the (hetero)aryl halides were evaluated by introducing electron-withdrawing and electron-donating groups on the aryl moiety. Although electron-poor (hetero)aryl halides (e.g., **3c**, **3d**, **3i**) underwent coupling faster than electron rich ones (**3a**, **3e**), the desired products were successfully obtained in all cases. Unfortunately, chlorobenzene couldn’t provide the product **3b**. By changing which heteroaryl chlorides were employed, couplings were smoothly proceeded, although the yield was decreased. A more sterically encumbered 2-methylimidazole also reacted without incidents (**3l**, **3n**).

In order to explore the feasibility of this approach, cyclic secondary amines (piperidine and pyrrolidine), acyclic secondary amine (diethylamine), aliphatic primary amine (ethanolamine) were examined to achieve the desired coupling (Table 3). First, the reactions were carried out by using 2-chloropyridine as starting material. To our delight, the corresponding coupling product **4a** and **4b** were obtained in 87% and 88% yield, respectively. We also attempted the coupling of 4-chloropyridine to form **4c** and **4d**. In both cases, the products were obtained in good yields. Both electron-rich (**4g**) and electron-poor (**4i**) aryl bromides participated equally well in the reaction. The reaction was effective in the presence of unprotected polar functional groups such as alcohol. It was encouraged that the ethanolamine substitute could provide **4l** in 90% yield; the lower yield was due to incomplete conversion of the starting material.

Although *N*-heterocycle electrophiles were the primary focus of this study, amino acids-based electrophiles were also evaluated (Table 4). Amino acids underwent coupling to afford corresponding products in moderate yields. Although the yields were modest, it was noted that these reactions were conducted under the conditions developed for the 2-bromoanisole with minimal reoptimization. Substrates bearing either electron-withdrawing (**5b**) or electron-donating groups (**5c**) on the (hetero)aryl halides coupled with high yields. Introduction of an ester group into amino acid was also tolerated (**5a**). Finally, we were pleased to find that our method was not limited to 2-chloropyrimidine. Using 2-chloropyrimidine as the substrate led to the formation of **5e**, **5f** in 86%, 87% yield, respectively.

The ligand BO has been reported to be used as a metal chelating agent [28], which is a typical *N*,*O*-ligand. Thus, it was believed that the Cu-catalyzed couplings could process in a homogeneous manner due to the formation of Cu-benzoinoxime complex. Herein, it was proposed that the possible mechanism for the couplings might run via a prototypical Cu (I)/Cu (III) catalytic cycle. [29] As shown in Figure 2, the catalytic cycle initiated from a Cu complex (**A**). Then, coordinated copper species (**B**) were produced via oxidative addition of an ArX and **A**. Ligand exchange was subsequently occurred between **A** and *N*-heterocycles to form intermediate **C,** which could be converted to **D** in the presence of base. The *N*-arylazole was obtained by final reductive elimination of **D**.

## 3. Materials and Methods

All of the starting materials, reagents, and solvents are commercially available and used without further purification. Melting points were determined with a X-4 apparatus (Beijing Taike Instrument Co., Ltd., Beijing, China) and were uncorrected. The nuclear magnetic resonance (NMR) spectra were recorded on a Bruker (Bruker Technology Co., Ltd., Karlsruhe, Germany) 400 MHz spectrometer in CDCl_3_ or DMSO-*d_6_* using tetramethylsilane (TMS) as an internal standard. Electrospray ionization mass spectrometry (MS (ESI)) analyses were recorded in an Agilent 1100 Series MSD Trap SL (Santa Clara, CA, USA). The reactions were monitored by thin-layer chromatography (TLC: HG/T2354-92, GF254), and compounds were visualized on TLC with UV light (Gongyi Yuhua Instrument Co., Ltd, Zhengzhou, China).

### General Procedure for Catalytic Experiments

To a solution of (hetero)aryl halide (2.81 mmol), *N*-nucleophile (3.37 mmol), BO (0.28 mmol), K_3_PO_4_ (5.62 mmol) in DMSO (4 mL), were added Cu(OAc)_2_ (0.28 mmol). The flask was evacuated and backfilled with argon for three times. The resulting suspension was heated in a 80 °C oil bath with stirring for the indicated time. The reactor was cooled to r.t., the flask was opened to air and the reaction mixture was poured into water (20 mL), extracted with ethyl acetate (20 mL × 3), and organic layer was washed with water (20 mL × 2) and once with brine (25 mL), dried over magnesium sulfate and concentrated *in vacuo*. The product was purified by column chromatography on silica gel using petroleum ether and ethyl acetate as eluent.

*1-(2-Methoxyphenyl)-1H-pyrrole (***3a***)* [30]: colorless oil (0.43 g, 88%). ^1^H-NMR (400 MHz, CDCl_3_) δ (ppm): 7.30–7.23 (2H, m), 7.03–6.98 (4H, m), 6.30 (2H, t, *J* = 2.2 Hz), 3.82 (3H, s). MS (ESI) *m*/*z*: 174.11 [M + H]^+^, see Appendix A.

*1-Phenyl-1H-pyrrole (***3b***)* [31]: white solid (0.38 g, 94%). m.p. 60–62 C. ^1^H-NMR (400 MHz, CDCl_3_) δ (ppm): 7.43–7.37 (4H, m), 7.25–7.21 (1H, m), 7.08 (2H, t, *J* = 2.2 Hz), 6.34 (2H, t, *J* = 2.2 Hz). MS (ESI) *m*/*z*: 144.04 [M + H]^+^.

*3-(1H-Pyrazol-1-yl)benzaldehyde (***3c***)* [32]: off white solid (0.46 g, 95%), m.p. 28–30 °C. ^1^H-NMR (400 MHz, CDCl_3_) δ (ppm): 10.08 (1H, s), 8.19 (1H, t, *J* = 1.8 Hz), 8.05–8.02 (2H, m), 7.81–7.76 (2H, m), 7.64 (1H, t, *J* = 7.9 Hz), 6.52 (1H, t, *J* = 2.1 Hz). MS (ESI) *m*/*z*: 173.08 [M + H]^+^.

*2-(1H-Pyrazol-1-yl)benzoic acid (***3d***)* [33]: white solid (0.51 g, 96%), m.p. 128–129 °C. ^1^H-NMR (400 MHz, CDCl_3_) δ (ppm): 11.40 (1H, br), 8.05–8.02 (1H, dd, *J* = 7.8 Hz, 1.2 Hz), 7.76–7.74 (2H, m), 7.62–7.58 (1H, m), 7.49–7.40 (2H, m), 6.48 (1H, s). MS (ESI) *m*/*z*: 189.06 [M + H]^+^.

*2-(1H-Pyrazol-1-yl)aniline (***3e***)* [34]: brown oil (0.38 g, 85%). ^1^H-NMR (400 MHz, CDCl_3_) δ (ppm): 7.74–7.71 (2H, m), 7.19–7.13 (2H, m), 6.85–6.76 (2H, m), 6.44 (1H, t, *J* = 2.0 Hz), 4.63 (2H, br). MS (ESI) *m*/*z*: 160.09 [M + H]^+^.

*1-(4-(1H-Pyrazol-1-yl)phenyl)ethan-1-one (***3f***)* [35]: yellow oil (0.47 g, 90%). ^1^H-NMR (400 MHz, CDCl_3_) δ (ppm): 8.08–8.06 (2H, m), 8.02 (1H, d, *J* = 2.5 Hz), 7.83–7.81 (2H, m), 7.78 (1H, d, *J* = 1.4 Hz), 6.53 (1H, t, *J* = 2.0 Hz), 2.63 (3H, s). MS (ESI) *m*/*z*: 187.09 [M + H]^+^.

*Ethyl 3-(1H-pyrazol-1-yl)benzoate (***3g***)* [36]: yellow liquid (0.55 g, 91%). ^1^H-NMR (400 MHz, CDCl_3_) δ (ppm): 8.31 (1H, t, *J* = 1.8 Hz), 7.99 (1H, d, *J* = 2.4 Hz), 7.97–7.93 (2H, m), 7.74 (1H, d, *J* = 1.6 Hz), 7.53 (1H, t, *J* = 8.0 Hz), 6.49 (1H, t, *J* = 2.0 Hz), 4.44–4.38 (2H, q, *J* = 14.3 Hz, 7.2 Hz), 1.41 (3H, t, *J* = 7.1 Hz). MS (ESI) *m*/*z*: 217.10 [M + H]^+^.

*Methyl 4-(1H-pyrazol-1-yl)benzoate (***3h***)* [37]: white solid (0.53 g, 93%), m.p. 103–105 °C. ^1^H-NMR (400 MHz, CDCl_3_) δ (ppm): 8.15–8.12 (2H, m), 8.00 (1H, d, *J* = 2.4 Hz), 7.80–7.77 (2H, m), 7.76 (1H, d, *J* = 1.4 Hz), 6.51 (1H, t, *J* = 2.1 Hz), 3.93 (3H, s). MS (ESI) *m*/*z*: 203.11 [M + H]^+^.

*2-(1H-Pyrazol-1-yl)benzonitrile (***3i***)* [38]: yellow oil (0.47 g, 98%). ^1^H-NMR (400 MHz, CDCl_3_) δ (ppm): 8.15 (1H, d, *J* = 2.5 Hz), 7.81–7.79 (3H, m), 7.77 (1H, d, *J* = 1.3 Hz), 7.72–7.68 (1H, m), 6.55 (1H, t, *J* = 2.1 Hz). MS (ESI) *m*/*z*: 170.07 [M + H]^+^.

*2-(1H-Pyrrol-1-yl)pyridine (***3j***)* [39]: colorless oil (0.34 g, 84%). ^1^H-NMR (400 MHz, CDCl_3_) δ (ppm): 8.43–8.42 (1H, m), 7.75–7.71 (1H, m), 7.52 (2H, t, *J* = 2.3 Hz), 7.32 (1H, d, *J* = 8.3 Hz), 7.11–7.08 (1H, m), 6.36 (2H, t, *J* = 2.3 Hz). MS (ESI) *m*/*z*: 145.04 [M + H]^+^.

*2-(1H-Imidazol-1-yl)pyridine (***3k***)* [40]: white solid (0.36 g, 88%), m.p. 40–41 °C. ^1^H-NMR (400 MHz, CDCl_3_) δ (ppm): 8.50–8.48 (1H, m), 8.35 (1H, s), 7.85–7.80 (1H, m), 7.65 (1H, t, *J* = 1.3 Hz), 7.37–7.35 (1H, m), 7.27–7.23 (1H, m), 7.20 (1H, s). MS (ESI) *m*/*z*: 146.08 [M + H]^+^.

*2-(2-Methyl-1H-imidazol-1-yl)pyridine (***3l***)* [41]: colorless oil (0.38 g, 85%). ^1^H-NMR (400 MHz, CDCl_3_) δ (ppm): 8.56–8.54 (1H, m), 7.86–7.82 (1H, m), 7.32–7.29 (2H, m), 7.28 (1H, d, *J* = 1.5 Hz), 7.02 (1H, d, *J* = 1.5 Hz), 7.02 (1H, d, *J* = 1.4 Hz), 2.59 (3H, s). MS (ESI) *m*/*z*: 160.09 [M + H]^+^.

*2-(1H-Imidazol-1-yl)pyrimidine (***3m***)* [42]: white solid (0.34 g, 83%), m.p. 120–122 °C. ^1^H-NMR (400 MHz, CDCl_3_) δ (ppm): 8.71 (2H, d, *J* = 4.8 Hz), 8.64 (1H, s), 7.90 (1H, s), 7.21 (1H, t, *J* = 4.8 Hz), 7.18 (1H, s). MS (ESI) *m*/*z*: 147.06 [M + H]^+^.

*2-(2-Methyl-1H-imidazol-1-yl)pyrimidine (***3n***)* [43]: white solid (0.36 g, 80%), m.p. 90–92 °C.^1^H-NMR (400 MHz, CDCl_3_) δ (ppm): 8.72 (2H, d, *J* = 4.8 Hz), 7.86 (1H, d, *J* = 1.4 Hz), 7.18 (1H, t, *J* = 4.8 Hz), 6.97 (1H, d, *J* = 1.3 Hz), 2.82 (3H, s). MS (ESI) *m*/*z*: 161.10 [M + H]^+^.

*2-(Piperidin-1-yl)pyridine (***4a***)* [44]: colorless oil (0.40 g, 87%). ^1^H-NMR (400 MHz, CDCl_3_) δ (ppm): 8.16–8.15 (1H, m), 7.44–7.39 (1H, m), 6.63 (1H, d, *J* = 8.6 Hz), 6.54–6.51 (1H, m), 3.52 (4H, d, *J* = 4.9 Hz), 1.62 (6H, s). MS (ESI) *m*/*z*: 163.13 [M + H]^+^.

*2-(Pyrrolidin-1-yl)pyridine (***4b***)* [45]: colorless oil (0.37 g, 88%). ^1^H-NMR (400 MHz, CDCl_3_) δ (ppm): 8.16–8.14 (1H, m), 7.44–7.39 (1H, m), 6.51–6.48 (1H, m), 6.35 (1H, d, *J* = 8.5 Hz), 3.46–3.43 (4H, m), 2.02–1.98 (4H, m). MS (ESI) *m*/*z*: 149.12 [M + H]^+^.

*4-(Piperidin-1-yl)pyridine (***4c***)* [46]: colorless oil (0.39 g, 85%). ^1^H-NMR (400 MHz, CDCl_3_) δ (ppm): 8.22–8.20 (2H, q, *J* = 5.1 Hz, 1.6 Hz), 6.62–6.61 (2H, q, *J* = 5.1 Hz, 1.6 Hz), 3.31 (4H, d, *J* = 4.9 Hz), 1.63 (6H, s). MS (ESI) *m*/*z*: 163.13 [M + H]^+^.

*4-(Pyrrolidin-1-yl)pyridine (***4d***)* [44]: colorless oil (0.36 g, 86%). ^1^H-NMR (400 MHz, CDCl_3_) δ (ppm): 8.18 (2H, d, *J* = 4.9 Hz), 6.36 (2H, d, *J* = 4.9 Hz), 3.30–3.27 (4H, m), 2.03–1.99 (4H, m). MS (ESI) *m*/*z*: 149.09 [M + H]^+^.

*2-(Pyrrolidin-1-yl)pyrimidine (***4e***)* [47]: colorless oil (0.37 g, 88%). ^1^H-NMR (400 MHz, CDCl_3_) δ (ppm): 8.32 (2H, d, *J* = 4.8 Hz), 6.45 (1H, t, *J* = 4.8 Hz), 3.59–3.56 (4H, m), 2.02–1.98 (4H, m). MS (ESI) *m*/*z*: 150.11 [M + H]^+^.

*2-(Piperidin-1-yl)pyrimidine* (**4f**) [47]: colorless oil (0.40 g, 87%). ^1^H-NMR (400 MHz, CDCl_3_) δ (ppm): 8.30 (2H, t, *J* = 5.6 Hz), 6.42 (1H, t, *J* = 5.6 Hz), 3.81–3.78 (4H, m), 1.72–1.59 (6H, m). MS (ESI) *m*/*z*: 164.14 [M + H]^+^.

*1-(2-Methoxyphenyl)pyrrolidine (***4g***)* [48]: colorless oil (0.40 g, 80%). ^1^H-NMR (400 MHz, CDCl_3_) δ (ppm): 6.90–6.81 (4H, m), 3.83 (3H, s), 3.30–3.27 (4H, m), 1.95–1.91 (4H, m). MS (ESI) *m*/*z*: 178.16 [M + H]^+^.

*1-(2-Methoxyphenyl)piperidine (***4h***)* [49]: colorless oil (0.44 g, 82%). ^1^H-NMR (400 MHz, CDCl_3_) δ (ppm): 6.99–6.84 (4H, m), 3.86 (3H, s), 2.99–2.97 (4H, m), 1.78–1.54 (6H, m). MS (ESI) *m*/*z*: 192.17 [M + H]^+^.

*4-(Pyrrolidin-1-yl)benzaldehyde (***4i***)* [50]: white solid (0.44 g, 89%), m.p. 83–85 °C.^1^H-NMR (400 MHz, CDCl_3_) δ (ppm): 9.72 (1H, s), 7.73 (2H, d, *J* = 8.8 Hz), 6.58 (2H, d, *J* = 8.8 Hz), 3.40–3.37 (4H, m), 2.08–2.02 (4H, m). MS (ESI) *m*/*z*: 176.13 [M + H]^+^.

*4-(Piperidin-1-yl)benzaldehyde (***4j***)* [51]: white solid (0.48 g, 90%). m.p. 63–64 °C. ^1^H-NMR (400 MHz, CDCl_3_) δ (ppm): 9.75 (1H, s), 7.75–7.71 (2H, m), 6.91 (2H, d, *J* = 8.9 Hz), 3.41–3.40 (4H, m), 1.68 (6H, s). MS (ESI) *m*/*z*: 190.16 [M + H]^+^.

*N,N-Diethylaniline (***4k**) [52]: yellow liquid (0.39 g, 92%). ^1^H-NMR (400 MHz, CDCl_3_) δ (ppm): 7.23–7.18 (2H, m), 6.69–6.61 (3H, m), 3.37–3.32 (4H, q, *J* = 14.1 Hz, 7.0 Hz), 1.15 (6H, t, *J* = 7.1 Hz). MS (ESI) *m*/*z*: 150.12 [M + H]^+^.

*2-((4-Nitrophenyl)amino)ethan-1-ol (***4l***)* [53]: yellow solid (0.46 g, 90%). m.p. 110–111 °C.^1^H-NMR (400 MHz, DMSO-*d_6_*) δ (ppm): 8.05 (2H, d, *J* = 9.2 Hz), 7.35 (1H, t, *J* = 5.0 Hz), 6.73 (2H, d, *J* = 9.2 Hz), 4.86 (1H, t, *J* = 5.4 Hz), 3.65–3.61 (2H, q, *J* = 11.2 Hz, 5.6 Hz), 3.31–3.27 (2H, q, *J* = 11.4 Hz, 5.7 Hz). MS (ESI) *m*/*z*: 183.08 [M + H]^+^.

*tert-Butyl phenyl-D-valinate (***5a***)* [54]: white solid (0.63 g, 90%), m.p. 64–66 °C. ^1^H-NMR (400 MHz, CDCl_3_) δ (ppm): 7.17–7.13 (2H, m), 6.72–6.69 (1H, m), 6.64–6.62 (2H, m), 4.12 (1H, br), 3.75 (1H, d, *J* = 5.3 Hz), 2.15–2.04 (1H, m), 1.42 (9H, s), 1.05–1.01 (6H, m). MS (ESI) *m*/*z*: 250.16 [M + H]^+^.

*(4-Nitrophenyl)glycine (***5b***)* [55]: brown solid (0.51 g, 92%), 224–226 °C. ^1^H-NMR (400 MHz, DMSO-*d_6_*) δ (ppm): 8.02 (2H, d, *J* = 9.0 Hz), 7.44 (1H, t, *J* = 5.6 Hz), 6.66 (2H, d, *J* = 9.1 Hz), 3.98 (2H, d, *J* = 6.0 Hz). MS (ESI) *m*/*z*: 197.08 [M + H]^+^.

*Phenyl-D-phenylalanine (***5c***)* [56]: white solid (0.60 g, 88%), m.p. 173–176 °C ^.1^H-NMR (400 MHz, CDCl_3_) δ (ppm): 7.33–7.17 (7H, m), 6.80 (1H, t, *J* = 7.3 Hz), 6.62 (2H, d, *J* = 7.8 Hz), 4.32 (1H, t, *J* = 5.8 Hz), 3.30–3.10 (2H, m). MS (ESI) *m*/*z*: 242.11 [M + H]^+^.

*Phenyl-L-alanine (***5d***)* [57]: white solid (0.42 g, 90%), m.p. 133–135 °C. ^1^H-NMR (400 MHz, DMSO-*d_6_*) δ (ppm): 7.06 (2H, t, *J* = 7.8 Hz), 6.56–6.53 (3H, m), 3.95–3.89 (1H, q, *J* = 14.0 Hz, 7.0 Hz), 1.37 (3H, d, *J* = 7.0 Hz). MS (ESI) *m*/*z*: 166.07 [M + H]^+^.

*Pyrimidin-2-yl-D-valine (***5e***)*: white solid (0.47 g, 86%), 113–115 °C. ^1^H-NMR (400 MHz, CDCl_3_) δ (ppm): 11.24 (1H, br), 8.25 (2H, s), 7.22 (1H, d, *J* = 8.1 Hz), 6.55 (1H, t, *J* = 4.9 Hz), 4.62–4.59 (1H, q, *J* = 13.1 Hz, 5.0 Hz), 2.37–2.29 (1H, m), 1.06 (6H, t, *J* = 7.2 Hz). MS (ESI) *m*/*z*: 196.16 [M + H]^+^, 218.11 [M + H]^+^. ^13^ C-NMR (100 MHz, CDCl_3_) δ (ppm): 176.0, 161.2, 110.4, 59.4, 31.0, 18.9, 18.2.

*Pyrimidin-2-ylmethionine (***5f***)*: white solid (0.56 g, 87%). ^1^H-NMR (400 MHz, CDCl_3_) δ (ppm): 13.32 (1H, br), 8.29 (2H, br), 7.92 (1H, d, *J* = 6.6 Hz), 6.62 (1H, t, *J* = 4.9 Hz), 4.88–4.84 (1H, q, *J* = 12.2 Hz, 6.1 Hz), 2.70–2.64 (2H, m), 2.37–2.22 (2H, m), 2.11 (3H, s). MS (ESI) *m*/*z*: 228.13 [M + H]^+^. ^13^ C-NMR (100 MHz, CDCl_3_) δ (ppm): 175.8, 160.3, 110.3, 53.7, 31.9, 30.0, 15.4.

## 4. Conclusions

In summary, a highly effective coupling reaction has been developed for the preparation of *N*-aryl compounds. This transformation occurs with good to excellent yields. A variety of substituted (hetero)aryl halides can be used as electrophiles, and azoles, piperidine, pyrrolidine, and amino acids, etc. function as nucleophiles. The key to this discovery was the identification of benzoin oxime ligand that can promote the (hetero)aryl halides to the corresponding *N*-arylation compounds. Efforts to apply our Cu-based system to other catalytic reactions and to expand the scope of the *N*-nucleophiles to other classes of nucleophiles are currently underway in our laboratory.

## Figures and Tables

**Figure 1 molecules-24-04177-f001:**
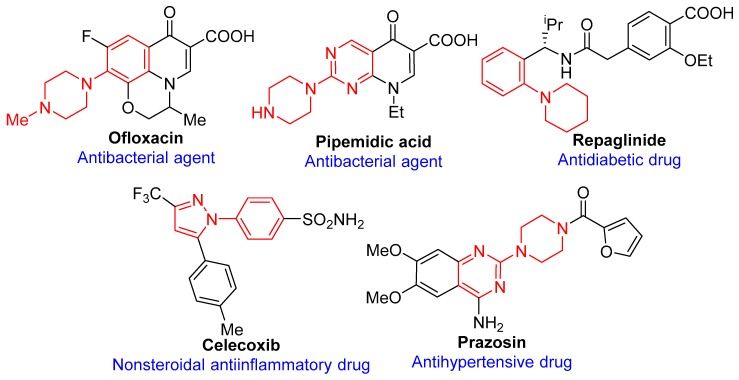
Chemical structures of selected pharmaceuticals containing the *N*-arylated core.

**Figure 2 molecules-24-04177-f002:**
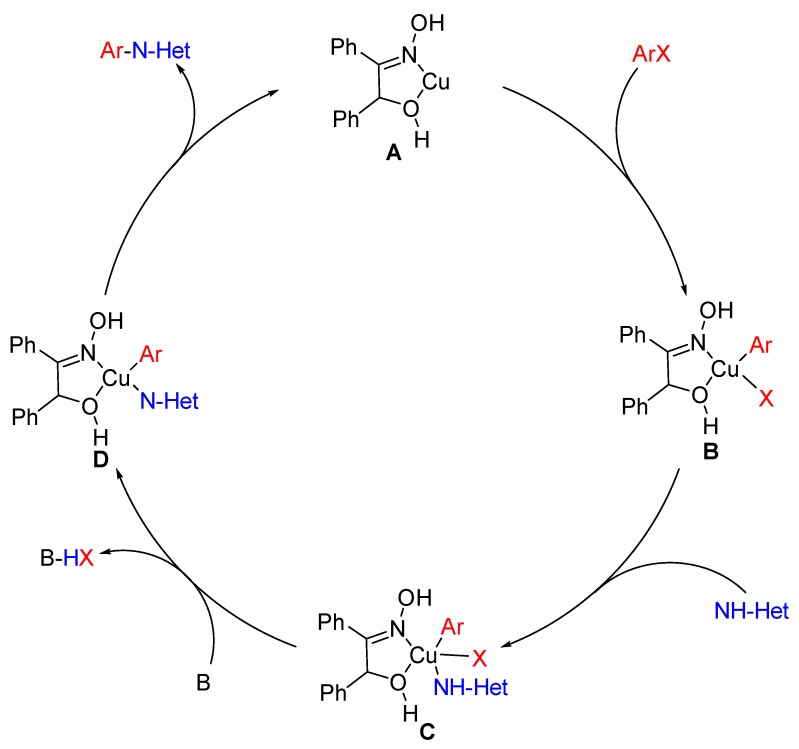
Proposed mechanism for the couplings of (hetero)aryl halides with *N*-containing heterocycles. NH-Het represented N-hetero nucleophiles; X was bromine or chloride.

**Table 1 molecules-24-04177-t001:**
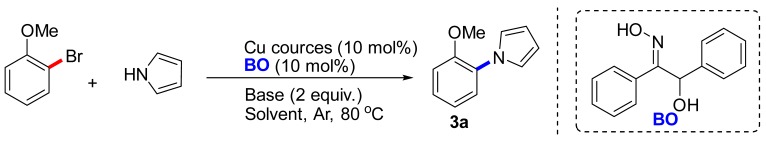
Identification of reaction condition ^a^.

Entry	Cu Sources	Base	Solvent	Yield (%) ^b^
1	CuI	K_2_CO_3_	DMF	12
2	Cu powder	K_2_CO_3_	DMF	15
3	Cu(OTf)_2_	K_2_CO_3_	DMF	22
4	Cu(II) gluconate	K_2_CO_3_	DMF	Trace
5	Cu_2_(OH)_2_CO_3_	K_2_CO_3_	DMF	Trace
6	Cu(OAc)_2_	K_2_CO_3_	DMF	53
7	Cu(OAc)_2_	Cs_2_CO_3_	DMF	60
8	Cu(OAc)_2_	K_3_PO_4_	DMF	72
9	Cu(OAc)_2_	NaHCO_3_	DMF	0 ^c^
10	Cu(OAc)_2_	Et_3_N	DMF	0 ^c^
11	Cu(OAc)_2_	*t*-BuOK	DMF	Trace
12	Cu(OAc)_2_	K_3_PO_4_	DMSO	90
13	Cu(OAc)_2_	K_3_PO_4_	Dioxane	43
14	Cu(OAc)_2_	K_3_PO_4_	DCE	Trace
15	Cu(OAc)_2_	K_3_PO_4_	Toluene	0 ^c^
16	Cu(OAc)_2_	K_3_PO_4_	H_2_O	0 ^c^
17	Cu(OAc)_2_	K_3_PO_4_	DMSO	70 ^d^
18	Cu(OAc)_2_	K_3_PO_4_	DMSO	15 ^e^
19	Cu(OAc)_2_	K_3_PO_4_	DMSO	90 ^f^
20	Cu(OAc)_2_	K_3_PO_4_	DMSO	61 ^g^
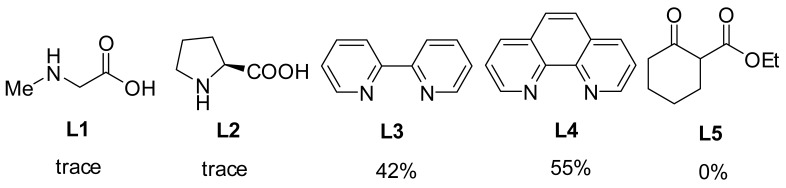

^a^ Reaction conditions: 2-Bromoanisole (2.81 mmol), pyrrole (3.37 mmol), Cu source (0.28 mmol), BO (0.28 mmol), solvent (4 mL), base (5.62 mmol), under Ar, at 80 °C, unless otherwise noted for 8 h. ^b^ Isolated yield with column chromatography. ^c^ Almost no reaction was observed by TLC. ^d^ 60 °C. ^e^ Without BO. ^f^ The loading of 2-bromoanisole was 28.1 mmol. ^g^ The reaction was carried out under air. Red bond indicated cleavage bond; blue bond indicated formed bond; and BO was α-benzoin oxime.

**Table 2 molecules-24-04177-t002:** C-N Coupling reactions of substituted aryl compounds with pyrrole or azoles ^a,b^.

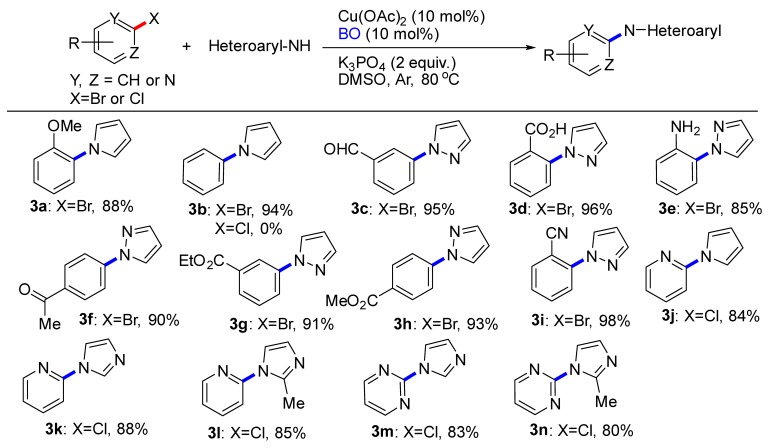

^a^ Reactions conducted on (hetero)aryl halides (2.81 mmol), pyrrole or azoles (3.37 mmol), Cu(OAc)_2_ (0.28 mmol), BO (0.28 mmol), K_3_PO_4_ (5.62 mmol), DMSO (4 mL) under Ar at 80 °C for 8~10 h. ^b^ Isolated yield with column chromatography.

**Table 3 molecules-24-04177-t003:** C-N Coupling reactions of substituted aryl compounds with amines ^a,b^.

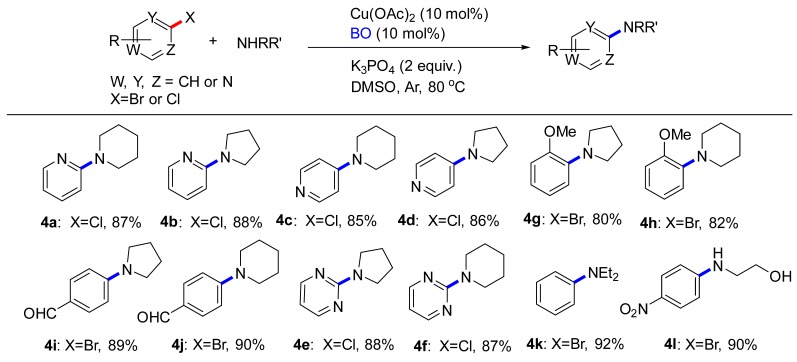

^a^ Reactions conducted on (hetero)aryl halides (2.81 mmol), amines (3.37 mmol), Cu(OAc)_2_ (0.28 mmol), BO (0.28 mmol), K_3_PO_4_ (5.62 mmol), DMSO (4 mL) under Ar at 80 °C for 8 h. ^b^ Isolated yield with column chromatography.

**Table 4 molecules-24-04177-t004:** C-N Coupling reactions of substituted aryl compounds with amino acids/esters ^a,b^.

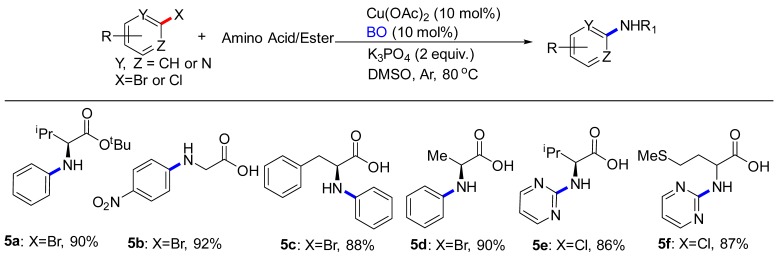

^a^ Reactions conducted on (hetero)aryl halides (2.81 mmol), amino acids/esters (3.37 mmol), Cu(OAc)_2_ (0.28 mmol), BO 0.28 mmol), K_3_PO_4_ (5.62 mmol), DMSO (4 mL) under Ar at 80 °C for 8 h. ^b^ Isolated yield with column chromatography.

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
