# Peer review of "Cu(II)-Catalyzed C-N Coupling of (Hetero)aryl Halides and N-Nucleophiles Promoted by α-Benzoin Oxime"

_molecules, 2019, doi:10.3390/molecules24224177_

Round 1

Reviewer 1 Report

Cu(II)-Catalyzed C-N Coupling of (Hetero)aryl Halides and N-Nucleophiles Promoted by α-Benzoin Oxime

The authors report on the copper catalysed C-N coupling of various (hetero)aryl halides and N-nucleophiles using a BO-Cu(II) catalytic system. The chemistry is on solid ground and the products/processes are well reported. The experimental data is exceptionally well done. This paper is well suited to be published in Molecules after these comments are addressed:

The authors discuss removal of the catalytic system via filtration but no details is given in the experimental section. Furthermore, no details are given on the actual state of the catalyst (is it still active? Can it be reused?). other than one phrase, the authors do not elaborate on this. If no convincing arguments are given for the need to filter the catalyst or to the actual experimental protocol and state of the catalyst after filtration, I suggest removing this part entirely from the paper.

I’ve noticed that only aryl bromides are used whilst pyridine-based halides can tolerate bromides and chlorides, is there a reason for this? Adding a phrase expanding on the reactivity (or lack of reactivity) of aryl chlorides would be very helpful for the scientific community.

Experimental data only mentions 1H NMR data for known compounds, which I have no problem with. 13C NMR would help cement the characterisation of these molecules (especially considering the title of the journal), but it’s not mandatory as far as I’m concerned since these are already known substrates. However, molecules 5e and 5f are not referenced and thus I presume they are not known; in these cases, where the compounds are not previously reported, 13C NMR must be done and reported.

On a more curious notes, how does the system tolerate air. If the reaction is done in air, what would be the yield?

There are quite a few mishaps with the English. Some phrases need to be corrected and/or polished.. for example: Abstract: Here is first reported the new application of a translate metal Page 2, line 43: However, it is never reported the use. Page 2, line 51: the catalyst is readily removal just by filtration Ce2CO3 should be Cs2CO3 Page 5, line 114: “universally” should be changed or removed -etc..

After these points are addressed, I recommend publication in Molecules.

Author Response

Dear reviewer,

Thank you for the time and effort that you have put into reviewing the previous version of the manuscript. The suggestions have enabled us to improve our work. Based on the instructions provided in your letter, we uploaded the file of the revised manuscript.

Appended to this letter is our point-by-point response to the comments raised by the reviewer 1. The comments are reproduced and our responses are given directly afterward.

Point 1: The authors discuss removal of the catalytic system via filtration but no details is given in the experimental section. Furthermore, no details are given on the actual state of the catalyst (is it still active? Can it be reused?). other than one phrase, the authors do not elaborate on this. If no convincing arguments are given for the need to filter the catalyst or to the actual experimental protocol and state of the catalyst after filtration, I suggest removing this part entirely from the paper.

Response 1: Thank you for the suggestion. It is just predicted by us that the catalyst is easily removal as the poor solubility of the Cu-ligand complex. However, there are not any convincing arguments of the actual experimental protocol and state of the catalyst. So, we have removed this part entirely according to this comment.

Point 2: I’ve noticed that only aryl bromides are used whilst pyridine-based halides can tolerate bromides and chlorides, is there a reason for this? Adding a phrase expanding on the reactivity (or lack of reactivity) of aryl chlorides would be very helpful for the scientific community.

Response 2: The coupling reaction of aryl chloride with pyrrole was suppled in table 2, but no product was produced, indicating that the inactive aryl chlorides are not suitable substrates. However, pyridine-based halides can tolerate chlorides. It is reasoned that the pyridine-based aryl ring is electron-deficient, which leading to the activity improvement of aryl chlorides.

Point 3: However, molecules 5e and 5f are not referenced and thus I presume they are not known; in these cases, where the compounds are not previously reported, 13C NMR must be done and reported.

Response 3: 13C NMR spectra of the new molecules 5e and 5f have been added in the manuscript and supporting information file.

Point 4: On a more curious notes, how does the system tolerate air. If the reaction is done in air, what would be the yield?

Response 4: According to your excellent suggestion, we performed the reaction under air, but a decreasing yield was observed. The result was added to the manuscript.

Point 5: There are quite a few mishaps with the English. Some phrases need to be corrected and/or polished.. for example: Abstract: Here is first reported the new application of a translate metal Page 2, line 43: However, it is never reported the use. Page 2, line 51: the catalyst is readily removal just by filtration Ce2CO3 should be Cs2CO3 Page 5, line 114: “universally” should be changed or removed -etc..

Response 5: A few mishaps with the English have been corrected under revision mode.

Finally, we appreciate very much for your time in editing our manuscript and the reviewer’s valuable suggestions and comments.

Sincerely,

Chunling Yuan

Reviewer 2 Report

Yuan et al. report on the use of the combination of Cu(II) sources with alfa-benzoin oxime as efficient catalytic system for the C-N coupling of heteroaryl halides and N-nucleophiles. Although the chemistry seems to have been done in a proper manner, the manuscript requires a thorough revision of English grammar by a native speaker, since the MS is at times hard to follow and particularly the abstract and the first paragraph of the results and discussion section is confusing, although the rest of the MS also requires to be fixed. Moreover, authors only make a mixture of reactants to prepare the catalyst and they must to go with this research further. That means to know what kind of complexes are they forming under the conditions they are using, to really clear whether is a copper complex with BO, or just a procedure to attain Cu(0) nanoparticles, and thus define whether the catalysts is homogeneous or heterogenous in nature. If it is homogenous a characterization of the catalysts or at least the catalyst precursor must be provided, together with a mechanistic proposal using this “complex”. On the other hand, if the catalysts is heterogeneous i.e. Cu(0) nanoparticles must be characterized. All this information will surely improve not only the quality of this MS but also the better comprehension of this interesting catalytic system. Finally, some key references are missing at the reference section and must be included before any further consideration of this MS, such as: (a) Inorg. Chim. Acta 363 (2010) 1262-1268; (b) A.S. Guram, R.A. Rennels, S.L. Buchwald, Angew. Chem.,Int. Ed. Engl. 34 (1995) 1348; (c) J. Louie, J.F. Hartwig, Tetrahedron Lett. 36 (21) (1995) 3609; (d) B.H. Yang, S.L. Buchwald, J. Organomet. Chem. 576 (1999) 125; (e) J.F. Hartwig, Angew. Chem., Int. Ed. 37 (1998) 2047; (f) S.L. Buchwald, Acc. Chem. Res. 31 (1998) 805.

Once this queries have been properly addressed the MS will require further revision.

Author Response

Dear reviewer,

We were pleased to know that our work was rated as potentially acceptable for publication in Molecules, subject to major revision. Thank you for the time and effort that you have put into reviewing the previous version of the manuscript. Your suggestions have enabled us to improve our work. Based on the instructions provided in your letter, we uploaded the file of the revised manuscript.

Appended to this letter is our point-by-point response to the comments raised by the reviewer 2. The comments are reproduced and our responses are given directly afterward.

Point 1: Although the chemistry seems to have been done in a proper manner, the manuscript requires a thorough revision of English grammar by a native speaker, since the MS is at times hard to follow and particularly the abstract and the first paragraph of the results and discussion section is confusing, although the rest of the MS also requires to be fixed.

Response 1: For the English grammar, we have checked the whole article again and again. Many grammar mistakes and description errors have been thoroughly modified under revision mode.

Point 2: authors only make a mixture of reactants to prepare the catalyst and they must to go with this research further. That means to know what kind of complexes are they forming under the conditions they are using, to really clear whether is a copper complex with BO, or just a procedure to attain Cu(0) nanoparticles, and thus define whether the catalysts is homogeneous or heterogenous in nature. If it is homogenous a characterization of the catalysts or at least the catalyst precursor must be provided, together with a mechanistic proposal using this “complex”. On the other hand, if the catalysts is heterogeneous i.e. Cu(0) nanoparticles must be characterized.

Response 2: The comments were highly appreciated. The ligand BO has been reported to be used as a metal chelating agent, which is a typical N,O-ligand. Thus, it was believed that the Cu-catalyzed couplings could process in a homogeneous manner due to the formation of Cu-benzoinoxime. Meanwhile, we also proposed the possible mechanism. These arguments have been added in the manuscript. Due to the limitations of analysis conditions, we are seeking cooperation with external scientific research platforms for structural analysis and mechanism verification of complex.

Point 3: some key references are missing at the reference section and must be included before any further consideration of this MS, such as: (a) Inorg. Chim. Acta 363 (2010) 1262-1268; (b) A.S. Guram, R.A. Rennels, S.L. Buchwald, Angew. Chem.,Int. Ed. Engl. 34 (1995) 1348; (c) J. Louie, J.F. Hartwig, Tetrahedron Lett. 36 (21) (1995) 3609; (d) B.H. Yang, S.L. Buchwald, J. Organomet. Chem. 576 (1999) 125; (e) J.F. Hartwig, Angew. Chem., Int. Ed. 37 (1998) 2047; (f) S.L. Buchwald, Acc. Chem. Res. 31 (1998) 805.

Response 3: Thank you for the suggestions, we accept the reviewer’s suggestion and have added these references about Pd-catalyzed C-N couplings in the first paragraph of the introduction part. And the cited references have also been renumbered.

Finally, we appreciate very much for your time in editing our manuscript and the reviewer’s valuable suggestions and comments.

Sincerely,

Chunling Yuan

Reviewer 3 Report

This paper reports C-N coupling of aryl halides with N-nucleophiles in the presence of Cu catalyst bearing a benzoin oxime ligand. The desired coupling products were successfully obtained in good yields with good functional-group compatibility. Various N-nucleophiles including heteroaromatic compounds and amino acids can be used. The results should contribute to further progress in practical copper catalysis. Thus, I think that this paper deserves publication in this journal, although some revisions are needed as shown below.

(1) Table 1, entry 7

Cs2CO3 instead of Ce2CO3 ?

(2) The key of this work is the use of the benzoin oxime ligand. To show the advantages of the benzoin oxime ligand, the results of other reported ligands, such as Ma’s ligand, should be shown as comparisons.

Author Response

Dear reviewer,

We are pleased to know that our work was rated as potentially acceptable for publication in Molecules. We thank you for the time and effort that you have put into reviewing the previous version of the manuscript. Your suggestions have enabled us to improve our work. Based on the instructions provided in your letter, we uploaded the file of the revised manuscript.

Appended to this letter is our point-by-point response to the comments raised by the reviewer 3. The comments are reproduced and our responses are given directly afterward.

Point 1: Table 1, entry 7 Cs2CO3 instead of Ce2CO3 ?

Response 1: The mistake has been corrected under revision mode.

Point 2: The key of this work is the use of the benzoin oxime ligand. To show the advantages of the benzoin oxime ligand, the results of other reported ligands, such as Ma’s ligand, should be shown as comparisons.

Response 2: These comments were highly appreciated. We accept the reviewer’s suggestion and have checked other ligands (including N,N-, N,O-, O,O-type bidentate ligands) that were applied in Cu-catalyzed N-arylation reactions. The results are shown in table 1 of the mauscript.

Finally, we appreciate very much for your time in editing our manuscript and the reviewer’s valuable suggestions and comments.

Best regards,

Chunling Yuan

Round 2

Reviewer 2 Report

Authors have complied properly with the queries requested. Thus, I believe that the MS can be accepted in its corrected version.

Reviewer 3 Report

The paper has been well revised. I recommend it for publication.